# Prognostic Significance of Ultrasound Characteristics and Body Mass Index in Patients with Apparent Early-Stage Cervical Cancer: A Single-Center, Retrospective, Cohort Study

**DOI:** 10.3390/diagnostics13040583

**Published:** 2023-02-04

**Authors:** Nicolò Bizzarri, Antonella Biscione, Francesca Moro, Luigi Pedone Anchora, Valeria Catinella, Camilla Certelli, Elena Teodorico, Anna Fagotti, Francesco Fanfani, Ali Kucukmetin, Denis Querleu, Gabriella Ferrandina, Giovanni Scambia, Antonia Carla Testa

**Affiliations:** 1UOC Ginecologia Oncologica, Dipartimento per la Salute della Donna e del Bambino e della Salute Pubblica, Fondazione Policlinico Universitario A. Gemelli, IRCCS, 00168 Rome, Italy; 2Università Cattolica del Sacro Cuore, 00168 Rome, Italy; 3Northern Gynaecological Oncology Centre, Queen Elizabeth Hospital, Gateshead NE9 6SX, UK

**Keywords:** cervical cancer, BMI, anthropometric characteristics, prognostic factors, ultrasonography, personalized medicine

## Abstract

The primary aim of the present study was to investigate the prognostic impact (defined as disease-free—DFS and overall survival—OS) of the ultrasound scan tumor parameters, patients’ anthropometric parameters, and their combination in early-stage cervical cancer. The secondary aim was to assess the relation between ultrasound characteristics and pathological parametrial infiltration. This is a retrospective, single-center, observational cohort study. Consecutive patients with clinical FIGO 2018 stage IA1–IB2 and IIA1 cervical cancer who underwent preoperative ultrasound examination and radical surgery between 02/2012 and 06/2019 were included. Patients who underwent neo-adjuvant treatment, fertility sparing surgery, and pre-operative conization were excluded. Data from 164 patients were analyzed. Body mass index (BMI) ≤20 Kg/m^2^ (*p* < 0.001) and ultrasound tumor volume (*p* = 0.038) were related to a higher risk of recurrence. The ratios between ultrasound tumor volume and BMI, ultrasound tumor volume and height, and ultrasound largest tumor diameter and BMI were significantly related to a higher risk of recurrence (*p* = 0.011, *p* = 0.031, and *p* = 0.017, respectively). The only anthropometric characteristic related to a higher risk of death was BMI ≤20 Kg/m^2^ (*p* = 0.021). In the multivariate analysis, the ratio between ultrasound-measured largest tumor diameter and cervix-fundus uterine diameter (with 37 as the cut-off) was significantly associated with pathological microscopic parametrial infiltration (*p* = 0.018). In conclusion, a low BMI was the most significant anthropometric biomarker impairing DFS and OS in patients with apparent early-stage cervical cancer. The ratios between ultrasound tumor volume and BMI, ultrasound tumor volume and height, and ultrasound largest tumor diameter and BMI significantly affected DFS but not OS. The ratio between ultrasound-measured largest tumor diameter and cervix-fundus uterine diameter was related to parametrial infiltration. These novel prognostic parameters may be useful in pre-operative workup for a patient-tailored treatment in early-stage cervical cancer.

## 1. Introduction

Despite that the introduction and implementation of cervical cancer screening and HPV vaccination have significantly reduced the incidence of cervical cancer in developed countries [1,2], cervical cancer represents the fourth most common cancer in women, with 604,127 new cases and 341,831 deaths worldwide in 2020 [3].

Tumor volume has been described as one of the major prognostic factors in early-stage cervical cancer [4,5]. There is much evidence in the literature reporting that ultrasound scan is accurate in detecting the tumor characteristics and in determining tumor volume, as well as for infiltration of peri-cervical tissues [6,7]. For this reason, recently, ultrasound has been included in the international guidelines of the European Society of Gynaecological Oncology (ESGO) for the assessment of pelvic tumor extent, in order to guide treatment options as an alternative to magnetic resonance imaging (MRI), when performed by a properly trained ultrasound examiner [8,9]. Patients’ anthropometric parameters have been poorly studied in relation to cervical cancer prognosis and clinico-pathologic characteristics. In this context, body mass index (BMI) has been shown to negatively influence cervical cancer patients’ survival, but this result is still controversial [10,11,12]. However, the relation between tumor dimensions and women’s anthropometric characteristics in patients with cervical cancer has not been previously investigated.

The rationale of the present study is represented by the fact that the combination of tumor in an ultrasound scan and the patient anthropometrics characteristics might have a major prognostic impact.

The primary aim of the present study was to investigate the prognostic impact (defined as disease-free—DFS and overall survival—OS) of the ultrasound scan tumor parameters, patients’ anthropometric parameters, and their combination in apparent early-stage cervical cancer. The secondary aim was to assess the correlation between ultrasound characteristics and pathological parametrial infiltration.

## 2. Materials and Methods

### 2.1. Study Design and Inclusion/Exclusion Criteria

This is a retrospective, single-center, observational cohort study, approved by the Institutional Review Board (number DIPUSVSP-27-07-2086 on 27 July 2020) at Fondazione Policlinico Universitario A. Gemelli, IRCCS (Rome, Italy). Consecutive patients with clinical International Federation of Gynecology and Obstetrics (FIGO) 2018 stage IA1–IB2 and IIA1 [13] cervical carcinoma, treated by primary radical surgical treatment with curative intent, between February 2012 and June 2019, at Fondazione Policlinico Agostino Gemelli IRCCS, Rome, Italy, were included. Only patients with available ultrasound examination performed within two weeks before surgery were included. Data were retrieved from the institution’s electronic database. Patients who underwent neo-adjuvant treatment, fertility sparing surgery, and pre-operative conization were excluded.

### 2.2. Ultrasound Scan

All patients underwent pre-operative ultrasound examination at our Ultrasound Center by a doctor with experience in gynecologic oncology ultrasound. Ultrasound examinations were performed with a transvaginal or transrectal probe to assess the pelvic organs. The methodology of cervical cancer assessment has been previously described [6,7,14]. High-end ultrasound equipment was used; the frequency of the vaginal probes varied between 5.0 and 9.0 MHz.

Tumor size was measured in three orthogonal diameters (craniocaudal, transverse, and anteroposterior). The craniocaudal extension was measured in a longitudinal section from the outermost margin of the tumor (facing the vagina) to the highest (most cranial) extension of the tumor (Figure 1), and the anteroposterior extension was measured perpendicular to this measurement. The transverse extension was measured in the transverse section between the outermost lateral margins of the tumor. The tumor volume was calculated with the formula of a sphere, Tvol = D1 × D2 × D3 × 0.53. Tumor echogenicity was classified as hypoechoic, isoechoic, hyperechoic, or mixed compared with the surrounding cervical tissue.

The tumors were assessed with color or power Doppler ultrasound. The intra-tumoral blood flow was classified subjectively as no detectable blood flow (color score = 1), minimal blood flow (color score = 2), moderate blood flow (color score = 3), or abundant blood flow (color score = 4) depending on the amount of color Doppler signals detected in the tumor [15].

### 2.3. Treatment

All patients underwent radical hysterectomy and bilateral pelvic lymphadenectomy with or without sentinel lymph node biopsy and salpingo-oophorectomy. Pre-operative staging was performed with a pelvic ultrasound scan, abdominal MRI-scan, and chest X-ray. The surgical approach was either laparotomy or minimally invasive (laparoscopy or robotic). The radicality of hysterectomy was classified according to the Querleu–Morrow classification [16]. In the case of suspicious initial parametrial involvement in the US-scan or MRI-scan, the patient was counseled for exclusive chemo-radiotherapy to be the first choice of treatment (as per ESGO guidelines [8]) or primary radical surgery, accepting the risk of potential adjuvant chemoradiation. A dedicated gynecologic oncology pathologist analyzed the surgical specimens. Adjuvant treatment was administered in line with international guidelines according to pathologic risk factors [8].

### 2.4. Study Variables

Information on ultrasound scan, anthropometric, clinical, and pathological characteristics was retrospectively collected.

Data were identified from the Red Cap (electronic data capture tools hosted at Fondazione Policlinico Universitario “A. Gemelli”, IRCCS; https://redcap-irccs.policlinicogemelli.it/) (last accessed 15 January 2023) database.

### 2.5. Statistical Analysis

Standard descriptive statistics were used to evaluate the distribution of each variable. Continuous variables were reported as median and categorical variables as frequency or percentage. DFS was defined as the time in months from the date of the surgery to the date of first recurrence, last follow-up, or death. OS was calculated as the time in months from the date of the surgery to the date of the last follow-up or death. OS and DFS were estimated by the Kaplan–Meier method [17] and the log-rank test was used to assess the statistical significance [18]. The relation between ultrasound and histology characteristics was assessed with logistic regression analysis. The influence of ultrasound and anthropometric parameters on their survival was analyzed using univariate and multivariate Cox proportional hazards models and described using hazard ratios (HRs) and their 95% confidence intervals [19]. Cut-offs for continuous variables in logistic and Cox regression analyses were determined with ROC curves according to the risk of recurrence.

The agreement between ultrasound tumor diameter and histological tumor diameter was assessed using Cohen’s kappa. Interpretation of Cohen’s index was performed according to Landis and Koch: ≤0.000 no agreement, 0.000–0.200 slight, 0.210–0.400 fair, 0.410–0.600 moderate, 0.610–0.800 substantial, and 0.810–1.000 almost perfect agreement [20].

All *p*-values reported are two-sided and a *p*-value <0.05 was considered statistically significant. 

The analysis was computed using SPSS version 26.0 (IBM Corporation 2018, Armonk, NY, USA).

## 3. Results

### 3.1. Patients’ Characteristics

Three hundred and eighty-three patients underwent surgery for newly diagnosed early-stage cervical cancer in the study period. Of these, 137 (35.8%) were excluded as they underwent conization before ultrasound and radical surgery, 46 (12.0%) were excluded as they underwent fertility sparing surgery, 18 (4.7%) were excluded as they did not undergo ultrasound, and 18 (4.7%) were excluded as there was no detectable tumor in the ultrasound. The exclusion process is presented in Figure 2. Therefore, 164 (42.8%) patients were included in the present study. Table 1 shows the characteristics of the entire population. The median age at diagnosis was 48 (range, 25–80) years. Most of the patients underwent minimally invasive radical hysterectomy (118/164, 72.0%), underwent type C radical hysterectomy (126/164, 76.8%), and were found to have squamous cell carcinoma at final histology (104/164, 63.4%). Most patients were FIGO stage IB1 (113/164, 68.9%). Twenty-seven (16.5%) had parametrial infiltration and 37/164 (22.5%) patients had metastatic lymph nodes at final histology.

The median largest tumor diameter measured at histology was 26.0 mm (range, 1.0–55.0). 

### 3.2. Ultrasound Scan Findings

Table 2 shows the ultrasound parameters of the study population. The median uterine volume was 74.1 mm^3^ (range, 11.9–966.7), the median largest tumor diameter was 29.0 mm (1.0–52.0), and the median tumor volume was 6.3 mm^3^ (range 1.0–58.5). Most cervical tumors appeared hypoechoic (122/164, 74.4%), with abundant color-Doppler vascularization (117/164, 71.3%) and with no internal uterine orifice involvement (104/164, 63.4%). 

In 15/164 (9.1%) patients, the ultrasound examiner suspected parametrial infiltration at the time of US scan.

### 3.3. Correlation between Ultrasound and Pathological Findings

The agreement between ultrasound tumor largest diameter and histological tumor diameter was substantial (Cohen’s kappa: 0.73, 95% CI: 0.625–0.801). 

The diagnostic performance in predicting parametrial infiltration with ultrasound was as follows: accuracy 79% (95% CI: 71.9%–84.9%), with 25/149 false-negative and 11/15 false-positive cases.

Univariate and multivariate logistic regression analyses relating ultrasound and histological tumor characteristics with pathologic parametrial infiltration are shown in Table 3. In the univariate analysis, the significant parameters were the ratio between largest tumor diameter and cervix-fundus uterine diameter (OR 3.551, 95%CI: 1.350–9.342) and lymph-vascular space involvement (LVSI) (OR 2.540, 95%CI: 1.008–6.400). In the multivariate analysis, the ratio between largest tumor diameter and cervix-fundus uterine diameter (with 37 as the cut-off) was significantly associated with pathological microscopic parametrial infiltration (OR 3.272, 95%CI: 1.231–8.702) (Table 3).

### 3.4. Survival Analysis

The median time of follow-up was 40 months (range, 3–94). Thirty-seven (22.6%) patients had recurrences and 12 (7.3%) died within the study period. Three-year DFS and OS were 83.7% and 92.1%, respectively. 

Three-year DFS in patients with BMI ≤20 Kg/m^2^ was 66.0% compared with 86.3% of patients with >20 Kg/m^2^ (*p* < 0.001). Three-year OS was 79.4% and 99.3% in the same groups of patients, respectively (*p* = 0.010) (Figure 3). Three-year DFS in patients with a ratio between ultrasound tumor volume and BMI of <65 and >65 was 87.3% and 70.6% (*p* = 0.027), respectively. Three-year OS in patients with a ratio between ultrasound tumor volume and BMI of <65 and >65 was 92.6% and 93.6% (*p* = 0.953), respectively (Figure 4).

Three-year DFS in patients with a ratio between ultrasound largest tumor diameter and BMI of <160 and >160 was 87.1% and 70.4% (*p* = 0.013), respectively. Three-year OS in patients with a ratio between ultrasound largest tumor diameter and BMI of <160 and >160 was 93.6% and 89.1% (*p* = 0.333), respectively (Figure 5).

Table 4 shows the Cox regression analysis for variables influencing the risk of recurrence and death. The variables significantly associated with a higher risk of recurrence were as follows: BMI (BMI ≤20 Kg/m^2^ at higher risk of recurrence, *p* < 0.001) and ultrasound tumor volume (*p* = 0.038). Similarly, the ratio between ultrasound tumor volume and BMI, the ratio between ultrasound tumor volume and height, and the ratio between ultrasound largest tumor diameter and BMI were significantly related to a higher risk of recurrence (*p* = 0.011, *p* = 0.031, and *p* = 0.017, respectively).

Regarding the risk of death, none of the ultrasound variables were related to the risk of death in the univariate Cox regression analysis (Table 4). The only anthropometric characteristic related to a higher risk of death was BMI ≤20 Kg/m^2^ (*p* = 0.021). 

Of note, the surgical approach (open versus minimally invasive) did not impact the risk of recurrence or death in the univariate Cox regression analysis (HR: 1.631, 95%CI: 0.631–4.216; *p* = 0.312 and HR: 1.624, 95%CI: 0.353–7.467; *p* = 0.534, for recurrence and death, respectively) [21].

## 4. Discussion

### 4.1. Main Findings

In the present study, we investigated the prognostic value of ultrasound tumor characteristics and anthropometric patients’ parameters in a series of patients with apparent early-stage cervical cancer. We found that BMI correlated with both the risk of recurrence and death, whereas the ratio between ultrasound parameters (such as tumor volume and largest diameter) and BMI only correlated with the risk of recurrence. Additionally, we also showed that pathological microscopic parametrial infiltration was related to the ratio between ultrasound largest tumor diameter and cervix-fundus uterine diameter. 

The present study aimed to assess the ultrasound characteristics in a population of early-stage cervical cancer patients and to evaluate whether there was a prognostic relation between tumor dimensions/volume detected at ultrasound and uterine dimensions or patient’s anthropometric characteristics. In other words, does the same tumor volume have a worse prognosis in a patient with a smaller uterus or a thinner patient? With our results, we demonstrated that BMI ≤20 Kg/m^2^ per se has a negative impact on DFS and OS; however, the ratios between ultrasound-measured tumor volume or between ultrasound-measured maximum tumor diameter and BMI also influenced DFS.

### 4.2. Comparison with Other Related Literature

Multiple studies in the literature have reported on the prognostic impact of largest tumor diameter and tumor volume on DFS and OS in cervical cancer [22,23]; similarly, a few reports have showed the prognostic impact of BMI, even though with different results [12,24]. In fact, some studies reported that only low BMI had a significant impact on survival [24], while others reported that the BMI extremes were related to a higher risk of death [12]. It is postulated that BMI may influence survival for the associated chronic systemic inflammation and poor nutritional status [25,26]. 

On the other side, the relation between tumor and uterine dimensions did not impact the survival, rejecting our initial hypothesis. Similarly, the internal uterine orifice and uterine isthmus involvement did not worsen the prognosis in our series. These results contrast with the current literature, which demonstrates that internal orifice involvement is an independent prognostic marker, in both early-stage and locally advanced cervical cancer [27,28]. With this background, we postulated not only that the cranio-caudal growth (toward uterine corpus) but also latero-lateral growth could have impacted the risk of recurrence, particularly in the case of a smaller uterus, where parametria and lymphatics are closer to the tumoral mass. Indeed, we realized that the ratio between largest tumor diameter and cervical-fundus diameter of the uterus was the factor most related to microscopic parametrial infiltration, which is a known negative prognostic marker. Further studies should focus on this aspect to predict parametrial infiltration and possibly avoid radical surgery in favor of exclusive chemoradiation [8].

### 4.3. Strengths and Limitations

The main limitation of the present study is represented by its retrospective nature and by the low number of underweight patients included. Nevertheless, to the best of our knowledge, this is the first study investigating the combination of ultrasound and anthropometric markers to predict pathologic (parametrial involvement) and oncological outcomes.

### 4.4. Future Perspectives 

The present results may represent a hypothesis-generating study that aims to emphasize the importance of pre-operative support of nutritional status before starting the treatment and to highlight the potential application of US to predict parametrial infiltration and the risk of recurrence.

Further research will be needed to validate the hypotheses of our study. Once confirmed, these data could promote research into novel prognostic factors and new treatment strategies.

## 5. Conclusions

In a population of patients with suspicious early-stage cervical cancer undergoing primary radical surgery, a low BMI was the most significant anthropometric biomarker impairing DFS and OS. The ratio between tumor volume measured in ultrasound and BMI and the ratio between largest tumor diameter measured in ultrasound and BMI significantly affected DFS but not OS. The pre-operative ratio between ultrasound-measured largest tumor diameter and cervix-fundus uterine diameter represented the factor most related to parametrial infiltration. These novel prognostic parameters might be useful in the pre-operative workup for a patient-tailored treatment in early-stage cervical cancer.

## Figures and Tables

**Figure 1 diagnostics-13-00583-f001:**
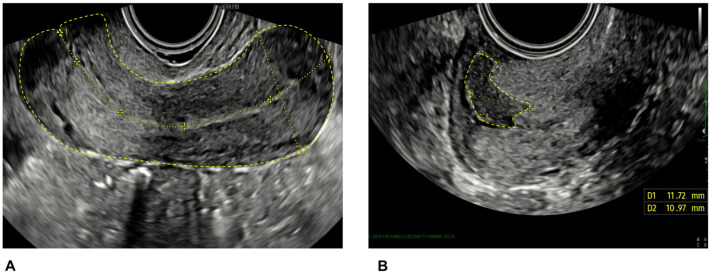
Ultrasound scan image showing the uterus (**A**) and tumor (**B**) diameters in two dimensions.

**Figure 2 diagnostics-13-00583-f002:**
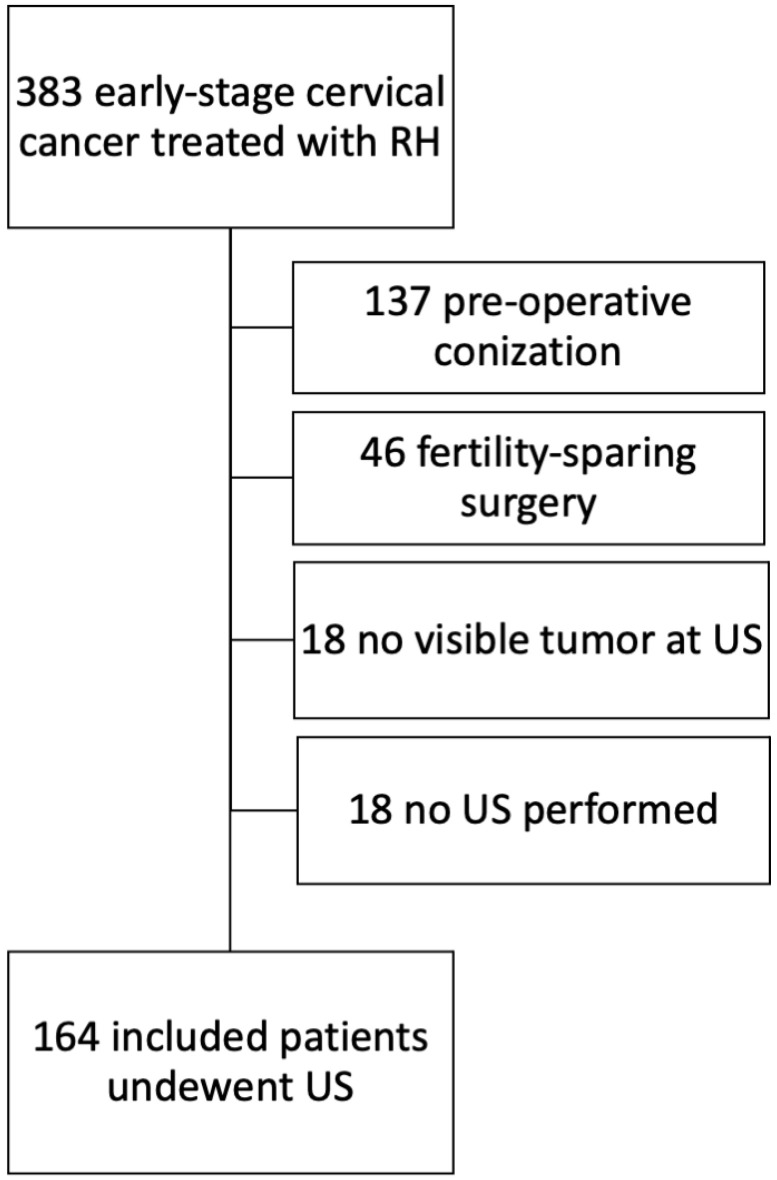
Flow chart demonstrating the exclusion process.

**Figure 3 diagnostics-13-00583-f003:**
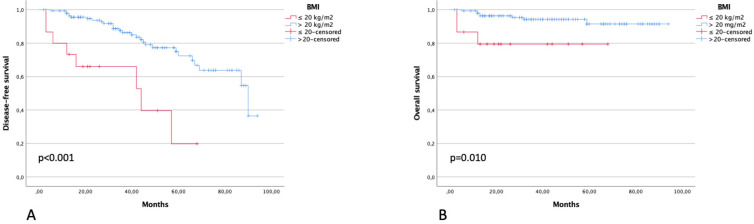
DFS (**A**) and OS (**B**) in patients with BMI of ≤20 and >20 kg/m^2^.

**Figure 4 diagnostics-13-00583-f004:**
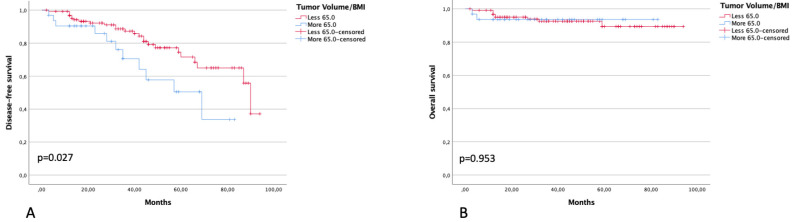
DFS (**A**) and OS (**B**) in patients with a ratio between tumor volume at US and BMI of ≤65 and >65.

**Figure 5 diagnostics-13-00583-f005:**
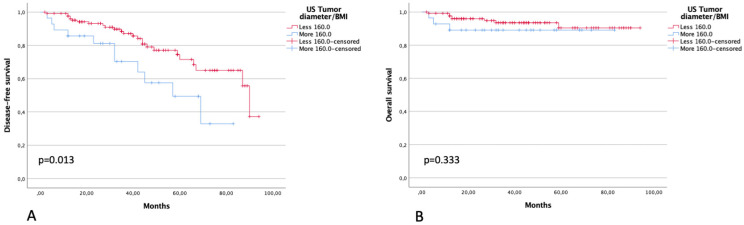
DFS (**A**) and OS (**B**) in patients with a ratio between largest tumor diameter at US and BMI of ≤160 and >160.

**Table 1 diagnostics-13-00583-t001:** Clinical, surgical, and histological characteristics of the study population.

Characteristics	*n* = 164 (Range, %)
**Age at diagnosis (years)**	48 (25–80)
**BMI ^a^ (kg/m^2^)**	24.0 (16.9–47.6)
**Height (cm)**	162 (150–180)
**Surgical Approach**	
Laparotomy	46 (28.0)
Laparoscopy	77 (47.0)
Robot	41 (25.0)
**Radicality of Radical Hysterectomy**	
A	2 (1.2)
B	36 (22.0)
C	126 (76.8)
**Histology**	
Squamous cell carcinoma	104 (63.4)
Adenocarcinoma	45 (27.4)
Adeno-squamous	5 (3.0)
Other	10 (6.1)
**Grade**	
1	3 (1.8)
2	89 (54.3)
3	70 (42.7)
Unknown	2 (1.2)
**Lymph-Vascular Space Infiltration**	
Negative	71 (43.3)
Positive	92 (56.1)
Unknown	1 (0.6)
**Pathologic Tumor Diameter**	
≤20 mm	46 (28.0)
>20 mm	114 (69.5)
Unknown	4 (2.4)
**Pathologic Depth of Stromal Infiltration (mm)**	8.2 (0.4–23.0)
**Largest Pathologic Tumor Diameter (mm)**	26.0 (1.0–55.0)
**Removed Lymph Nodes**	26 (1–75)
**Patients with Metastatic Lymph Nodes**	37 (22.5)
**Pathologic FIGO ^b^ Stage 2009**	
IA1	1 (0.6)
IB1	113 (68.9)
IB2	15 (9.1)
IIA1	7 (4.3)
IIA2	1 (0.6)
IIB	27 (16.5)

Results are presented as *n* (%) or median (range); ^a^ BMI: body mass index; ^b^ FIGO: International Federation of Gynecology and Obstetrics.

**Table 2 diagnostics-13-00583-t002:** Ultrasound characteristics of the study population.

Characteristics	*n* = 164 (Range, %)
Uterus volume (mm^3^)	74.1 (11.9–966.7)
Largest tumor diameter (mm)	29.0 (1.0–52.0)
Tumor volume (mm^3^)	6.3 (1.0–58.5)
Depth of cervical stromal infiltration	
<2/3	76 (56.3)
>2/3	88 (53.7)
Echogenicity	
Hypoechoic	122 (74.4)
Isoechoic	26 (15.9)
Hyperechoic	15 (9.1)
Not reported	1 (0.6)
Color-Doppler evaluation	
No detectable flow (color score 1)	5 (3.0)
Minimal flow (color score 2)	10 (6.1)
Moderate flow (color score 3)	32 (19.5)
Abundant flow (color score 4)	117 (71.3)
Internal uterine orifice involvement	
No	104 (63.4)
Yes	60 (36.6)

**Table 3 diagnostics-13-00583-t003:** Univariate and multivariate logistic regression analysis relating ultrasound and histological tumor characteristics with pathologic parametrial infiltration.

	Pathologic Parametrial Infiltration				
		Univariate Analysis		Multivariate Analysis	
Characteristics	Comparison/Measure	OR (95% CI)	*p*-Value	OR (95% CI)	*p*-Value
**Ultrasound**					
US stromal infiltration	Full thickness vs. superficial/middle third	1.242 (0.930–1.658)	0.142		
Largest tumor diameter (mm)	mm	1.000 (1.000–1.001)	0.415		
Echogenicity	Hypoechoic vs. non-hypoechoic	0.884 (0.358–2.178)	0.778		
US tumor maximum diameter/Uterine CF diameter	≤37 or >37	3.551 (1.350–9.342)	**0.010**	3.272 (1.231–8.702)	**0.018**
**Pathologic**					
LVSI	Negative vs. positive	2.540 (1.008–6.400)	**0.048**	2.203 (0.857–5.660)	0.101
Grade	1–2 vs. 3	1.348 (0.889–2.045)	0.160		
Pathologic tumor diameter	≤2 or >2 cm	2.234 (0.719–6.940)	0.165		

**Table 4 diagnostics-13-00583-t004:** Univariate Cox regression analysis for clinical and ultrasound variables influencing risk of recurrence and death*.

	**RISK OF RECURRENCE**		
**Characteristics**	**Comparison/Measure**	**HR (95% CI)**	** *p* ** **-Value**
Age	≤50 vs >50 years	0.608 (0.305–1.215)	0.159
BMI	≤20 vs >20 kg/m^2^	0.213 (0.095–0.474)	**<0.001**
Largest US Tumor diameter	mm	1.018 (0.983–1.054)	0.326
Depth of infiltration	Full thickness vs Superficial/Middle third	1.116 (0.899–1.387)	0.320
Echogenicity	Hypoechoic vs non-hypoechoic	1.023 (0.482–2.172)	0.953
Color-Doppler	Abundant flow vs non-abundant flow	1.096 (0.909–1.322)	0.337
Internal uterine os involvement	No vs Yes	0.863 (0.428–1.741)	0.681
Tumor Volume	mm^3^	1.035 (1.002–1.069)	**0.038**
Tumor Volume/BMI	continuous	1.010 (1.002–1.018)	**0.011**
Largest Tumor diameter/BMI	continuous	1.009 (1.002–1.017)	**0.017**
Tumor Volume/Height of patient	continuous	1.006 (1.001–1.011)	**0.031**
Largest Tumor diameter/Uterine CF diameter	continuous	1.003 (0.980–1.026)	0.830
Tumor CF diameter/Uterine CF diameter	continuous	0.995 (0.971–1.020)	0.673
	**RISK OF DEATH**		
**Characteristics**	**Comparison/Measure**	**HR (95% CI)**	** *p* ** **-Value**
Age	≤50 vs >50 years	0.455 (0.123–1.683)	0.238
BMI	≤20 vs >20 Kg/m^2^	0.205 (0.054–0.784)	**0.021**
Largest US Tumor diameter	mm	0.981 (0.925–1.041)	0.531
US depth of infiltration	Full thickness vs Superficial/Middle third	1.260 (0.844–1.881)	0.258
Echogenicity	Hypoechoic vs non-hypoechoic	3.505 (0.452–27.183)	0.230
Color-Doppler	Abundant flow vs non-abundant flow	0.976 (0.723–1.319)	0.877
Internal uterine orifice involvement	No vs Yes	1.041 (0.313–3.465)	0.948
Tumor Volume	mm^3^	0.975 (0.900–1.055)	0.527
Tumor Volume/BMI	continuous	0.998 (0.980–1.016)	0.824
Largest Tumor diameter/BMI	continuous	1.005 (0.991–1.018)	0.499
Tumor Volume/Height of patient	continuous	0.997 (0.985–1.010)	0.678
Largest Tumor diameter/Uterine CF diameter	continuous	0.992 (0.953–1.032)	0.685
Tumor CF diameter/Uterine CF diameter	continuous	0.968 (0.924–1.015)	0.176

* Multivariate analysis not performed for the risk of interaction between variables containing BMI. HR: hazard ratio; BMI: body mass index; CF: cervix-fundus; US: ultrasound scan.

## Data Availability

Data available upon reasonable request.

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
