# Peer review of "Prognostic Significance of Ultrasound Characteristics and Body Mass Index in Patients with Apparent Early-Stage Cervical Cancer: A Single-Center, Retrospective, Cohort Study"

_diagnostics, 2023, doi:10.3390/diagnostics13040583_

Round 1
Reviewer 1 Report
The manuscript "Prognostic significance of the interaction between ultrasound characteristics and body mass index in patients with cervical cancer treated with primary surgery: a single-center, retrospective, cohort study " is an interesting manuscript on the prognostic impact of the ultrasound scan tumour parameters, patients’ anthropometric parameters, and their combination, in early-stage cervical cancer. The work is original and well structured, giving important novelty to scientific literature. The design of the project is appropriate and the results are significant. The statistical analysis is well conducted and the language is acceptable. It represents a valid work and it is suitable for publication.
Author Response
We would like to thank the Reviewer for the nice comment. Regards.
Reviewer 2 Report
The title and abstract are clear. However, the title is too long, and if it is possible to make it more concise without losing its meaning, I would suggest doing it.
The introduction part is narrow and does not provide a full rationale for the study. Suggest making in more explicit, but focused at the same time.
Methods are well-described, however, the text required proper systematization. Please include the following and change some subheadings: study design, study subjects, exclusion/inclusion criteria, and study variables, (the other sections could be left as existing).
The results are supported by clear tables and figures.
Author Response
We thank the Reviewer for the interesting remarks.
As suggested, we have:
- Shortened the title
- Added a statement on the rationale of the study in the introduction
- We have included sub-headings in the methods
Thanks again.
Reviewer 3 Report
I read with great interest the Manuscript titled "Prognostic significance of the interaction between ultrasound characteristics and body mass index in patients with cervical cancer treated with primary surgery: a single-center, retrospective, cohort study" which falls within the aim of the Journal.
In my honest opinion, the topic is interesting enough to attract the readers’ attention. The abstract perfectly summarizes the contents of the manuscript. The introduction is satisfactory. Methodology is accurate and conclusions are supported by the data analysis. References are relevant to the research. The discussions sufficiently answer the following questions: main findings of the study, strength and Limitations of the study, implications and comparison with literature, future directions.
Considered all these points, I think it could be of interest for the readers and, in my opinion, it deserves the priority to be published.
Author Response
We would like to thank the Reviewer for the nice comment. Kind regards.